# Equitable Pathways to 2100: Professional Sustainability Credentials

**L. Julian Keniry**

ESDI Green Teach for Opportunity Project, Washington, DC 20007, USA; ljkeniry@earthlink.net;
Tel.: +1-202-999-9244

**Abstract:** Across numerous industries and occupations, professional associations are contributing to knowledge and skills for sustainability by offering new credentials. This represents an opportunity to increase students' career preparedness for clean economies that accomplish steep reductions in greenhouse gas emissions over the next thirty years. This also presents a particular opportunity to help lower-income young adults better position themselves for good jobs that make meaningful contributions to the societal transition ahead. Providing suggestions for navigating and embedding them into curricula, this article highlights seventeen sustainability credentials and mentions another fourteen. In addition to definitions, it also provides analysis of aspects such as third-party accreditation, student supports, academic and maintenance requirements, and fees. Internet research and e-mail correspondence with credentialed professionals was an iterative process in which the author set out with a list of aspects to consider, identified new aspects in the process of researching credentials, compared those aspects, and so on, in an iterative cycle of comparison across offerings. The result is both a representative list of non-academic, professional credentials worth consideration as complements to the higher education curriculum as well as a set of suggestions for engaging with them in ways that foster opportunity for students from all backgrounds.

**Keywords:** sustainability credentials; lower-skilled adults; middle-skilled adults; clean economy skills; corporate sustainability reporting; climate officers; sustainability officers; green building trades; sustainable buildings; sustainable materials; green supply chain; new careers

## 1. Introduction

When she declared at the UN climate meetings that adults should not expect children to create hope for the future, Greta Thunberg was right [1]. Indeed, it is the responsibility of adult educators and other community leaders to create viable pathways to contributions of every kind, from careers to callings, that can sustain families, communities, and the full diversity of life.

Nudging academic programs and businesses to redefine the knowledge, skills and overall competencies required across fields is one way to help create these new pathways. Creators of professional sustainability certifications aim to help do just that. By reducing barriers to entry for lower-skilled adults from diverse demographic backgrounds, post-secondary sustainability experience can yield powerful ancillary social benefits from stronger families to lower incarceration rates, and improved health [2].

A surprise for many educators when considering incorporating these credentials into academic programs may be just how many there are. Over the past ten years, some form of sustainability credential has emerged in nearly every field.

## 2. Materials and Methods

In this section, we explore the constellation of sustainability credentials as a whole. Parts 2.1 through 2.6.1 note the scope of the research, share roots of the idea to cover credentials and include research on benefits to students. They provide background on how examples and aspects were identified, sharing some of the observed trends along the way, such as the range of types of hosts and how the field is evolving. The last part (2.6.2) examines the link between equity and educational quality.

### 2.1. Research Scope

This paper highlights seventeen of these professional credentials for sustainability and mentions an additional fourteen. It weighs several indicators to help navigate this growing array of professional credentials for sustainability and provides tips for incorporating them as complements to general and more technically-oriented higher education programs. In addition to providing background on the variety and types of host organizations, it explores academic and social benefits, and examines quality variables and costs. Finally, the article addresses, directly and indirectly, the central question of this journal edition, which is how sustainability can enhance equity and leverage other social benefits. It does this primarily by identifying new channels for affordable and effective access to necessary competencies for sustainability that can enhance academic programming and career readiness. For lists of organizations, credentials and industry sectors, see Appendices A and B.

### 2.2. Roots

The realization that sustainability was redefining careers sprouted when I wrote *EcoDemia*, a book about leadership demonstrated by administrators and staff running operations as diverse as dining, purchasing and transportation [3]. It was clear that these roles complemented that of the formal curriculum. Faculty often seemed to appreciate this, too, as they asked staff to support the teaching in various ways. Walter Simpson, when he served as the energy officer for the University of Buffalo, for example, was asked to serve as guest lecturer for various classes and Kurt Teichert, sustainability officer at Brown University, occupied two desks—one in the physical plant and another in an academic department.

During later efforts to support career development across diverse fields by awarding fellowships and internships in the 2000s, student leaders helped to redefine whole fields from the building trades to finance. Many of these graduates today are leading careers related to their projects. Designing and implementing projects gave students precious insights into solutions for sustainability, but these experiences were often separate from their academic studies.

Two national surveys of approximately one thousand colleges and universities led by my team and me between 1999 and 2008 with National Wildlife Federation (NWF) and Princeton Survey Research Associates International indicated that few programs had infused sustainability across traditional curricula [4]. Although many hundreds of institutions across the world now offer certificate and degree programs for sustainability, according to the Association for the Advancement of Sustainability in Higher Education (AASHE) [5], infusion of new content and pedagogy for sustainability into traditional curricula is uncertain, as the above studies have yet to be repeated.

Professional associations, it seemed, could provide a complement to the curriculum and a bridge to careers, making competencies for sustainability more accessible to students while building new career connections and insights. Interest in this idea grew while organizing career development programming with community college teachers across the US participating in the Greenforce Initiative of Jobs for the Future and NWF between 2010 and 2016. Defining skills and competencies for sustainability was a key focus of the initiative and culminated in the *Greenprint* in 2016 [6], a guide to sustainability career skills developed with stakeholders from multiple sectors and trades across the US and Canada.

Our team responded by creating a credentialing program, called NWF EcoLeaders, along with a career center for undergraduate and graduate students that offers an annual careers conference. When

developing content and recruiting speakers for the first conference in 2017 [7], it became apparent that the well-known Leadership in Energy and Environmental Design Accredited Professional (LEED-AP) designations of the U.S. Green Building Council and its partner, the Green Building Certification Inc. (GBCI), were in company with more than two dozen other professional accreditations for sustainability offered by many organizations. Thus, the idea for more research was hatched.

## *2.3. Breadth of Concept*

"Sustainability" is used liberally here, in the context of professional credentials, to reflect an amalgam of common definitions, including that of the Brundtland Commission which states that "Humanity has the ability to make development sustainable to ensure that it meets the needs of the present without compromising the ability of future generations to meet their own needs." [8]. Also encompassed is the centrality of restoring biodiversity, curbing pollution and other adverse public health impacts, while advancing solutions that restore the overall environment, are economically savvy and socially responsible. Most of the professional credentials listed here are more heavily weighted in one or more of these areas, yet they address all of them to varying degrees.

## *2.4. Benefits to Students*

One social dimension of professional credentials for sustainability is the advantage they can give to lower-skilled adults by offering an avenue for post-secondary education in science, technology, engineering and math (STEM) and, thus, to higher wages. The Brookings Institution reports that 86 percent of STEM jobs require a post-secondary certification, AA degree or higher and pay more than jobs with similar academic requirements [9]. A later Brookings study finds that typical wages in the clean economy exceed those in the aggregate economy by more than 10 percent [10].

These credentials can also enhance college and university coursework. John Kanter, who has served as a wildlife specialist for the Cooperative Extension at the University of New Hampshire and as Senior Wildlife Biologist at the National Wildlife Federation, notes that he has "often encouraged students to pursue professional credentials through The Wildlife Society because they add perspective, help students make connections, and help them perform better academically." [11].

## *2.5. Categorizing Credentials*

Credentialing is becoming big business, and the term broadly encompasses many kinds of recognition. The sustainability portion of the realm is becoming substantial in number and scope of offerings, but it is still small compared with the number of credentials offered overall in the US and beyond. A project called the Credential Engine has identified nearly 740,000 badges, degrees, certificates, licenses, and diplomas and approximately half of those, reports Rebecca Koenig, are offered by non-academic organizations [12].

### 2.5.1. Four Groups

One way to organize the array of sustainability designations is to group them into four categories: activities, goods, people, and places. While this article features professional recognition of people, certifications focused on activities, goods, and places offer educational benefits as well. Many vibrant associations for sustainability that offer certifications do not credential people, but they do help participants develop important competencies and connections for sustainability. Examples of programs that do not yet credential people include the awards from the Sustainable Purchasing Leadership Council (SPLC), the Association for the Advancement of Sustainability in Higher Education (AASHE) Sustainability Tracking, Assessment and Rating System (STARs), and the Forest Stewardship Council (FSC) certification for forest products.

### 2.5.2. Identifying Examples and Aspects

Surveying professional credentials with potential to enhance post-secondary education and degree programs has been an iterative process. It has involved developing an initial list of known examples and comparative aspects, extensive online research within and across sectors, and editing the list of aspects as new examples and angles emerged. Other online research included reviewing recipient biographies and lists of member organizations, reading job descriptions and articles on the topic, discussions by email and in person with recipients, and vetting ideas with leading sustainability curriculum practitioners [13]. Winnowing a larger list of options was more feasible as comparative indicators of quality and other differentiating factors became more apparent.

### 2.5.3. Host Variety

While the featured hosts share a common impetus to transform practices for sustainability in their sectors, they vary in other ways. In some cases, associations that have existed for more than three decades, such as the Association of Energy Engineers (AEE) and the APICS Association for Supply Chain Managers (ASCM), have incorporated new competencies. They have either added an array of sustainability-focused credentials, in the former example, or incorporated sustainability aspects into existing credentials, in the latter.

Other credentialing host varieties include entrepreneurs, women-owned businesses and non-profits that have created new fields or tracks within existing fields. The International Society for Sustainability Professionals (ISSP) certificate fits the latter example of a newer organization filling an unmet niche and the Global Sustainability Institute of Technology (GSIT)'s Certified Green Supply Chain Professional (GSCP) designation falls into the former category of a long-standing organization adding new competencies. Other formats include universities in partnership with a professional society or other non-profit (e.g., biomimicry specialist) or long-standing associations that incorporated some sustainability facets (GSCP) or incorporated it to a more significant degree (AEE). See Appendix B for a list of credentialing organizations for sustainability by industry sector.

### 2.5.4. Making Room for New Ideas

Collectively, the credentialing organizations illustrate a kind of academic and occupational disruption for sustainability. One of the best examples of this is the U.S. Green Businesses Council (USGBC) and its Leadership in Energy and Environmental Design (LEED) certifications for buildings and credentials for people. These programs now recognize more than 96,000 registered and certified buildings across 167 countries [14] and more than 200,000 credential holders in the architecture, design, construction and allied fields [15]. The annual Greenbuild International Conference and Expo attracts upwards of 20,000 attendees.

### 2.6. An Evolving Field

For all of their innovations and strengths, however, sustainability credentialing reflects an evolving field. Among established industry associations, definitions and norms appear to be relatively more well developed. One reason may be that, while standards have been in place for auditing environmental management systems and energy efficiency for many years, they are new or still emerging for green purchasing, carbon accounting, finance and other professionals. Another reason may be limited knowledge or capacity to participate in the accreditation and standards industry.

A mapping program and other resources offered by the Institute for Credentialing Excellence helps executives identify where they are on the path to implementing various types and levels of recognitions for people. The program also helps practitioners choose which type of credential is most appropriate to offer. For example, while the purpose of assessment-based certificates, the Institute explains, is to "build capacity and recognition of a specialty area of practice or a set of skills," the

purpose of professional or personnel certification programs is to "assess knowledge, skills and or competencies previously acquired." [16].

### 2.6.1. Social Dimensions

Even when they do reflect best practices for professional accreditation, incorporation of social dimensions varies. Since most of the credentials featured here originated in Western, developed countries, even when they pay a nod to less developed regions or to indigenous peoples, they are formulated through a particular lens. Host associations do not always reflect diverse memberships or boards or explicitly mention diversity recruitment initiatives.

Then again, even credentials that do not at first glance appear to encompass social dimensions may address these topics in various ways. The Certified Wildlife Biologist designation of The Wildlife Society (TWS), for example, appears to be limited to the biological sciences, but its certification outlines address competing land use interests, ethics concerns and diversity. TWS and several of the other featured organizations have also kept fees relatively low and provided incentives for students from under-represented backgrounds, helping those who would otherwise have limited capacity to pursue post-secondary education.

### 2.6.2. Other Equity Angles

Social dimensions are important in sustainability, as they are in education. One of the reasons "sustainability" is often used in place of the term "environmental" is to better convey incorporation of social sciences along with natural sciences into decision-making. The idea of a triple bottom line, coined by John Elkington [17], is a way to emphasize that societal imperatives—such as clean air and water, healthy small businesses, and good medical care—are better accomplished when policies are mutually beneficial.

In education, the benefits of seeking this triple bottom line are considerable. The highest performing education systems across the world, finds Mel Ainscow at the University of Manchester, combine quality with equity. These programs ensure that social circumstances such as gender, ethnic origin or family background are not obstacles to achieving educational potential [18]. However, anthropologist and diversity scholar, Yolanda Moses, notes that the US "stands at a critical juncture concerning its ability to engage both the challenge and the promise of what it means to be an inclusive and engaged pluralistic democracy" [19].

If sustainability can help bridge the gap for lower-skilled adults and people of color, all of society could benefit from higher employment, educational attainment levels and more. The Georgetown Public Policy Institute finds, for example, that after the Great Recession in 2008, African Americans were twice as likely as Whites to be unemployed in the US. As jobs have recovered, meanwhile, the highest growth in opportunity has been for those with Bachelor's degrees, followed by those with Associate's degrees or some college.

Georgetown also reports that people who have post-secondary education and degrees in today's economy are earning seventy-four percent more than workers with a high school diploma or less. Yet, the US has been underproducing workers with post-secondary education [20]. Of those who do enroll in college, the Hechinger Report finds, enrollment rates are lower for Hispanics and Blacks than for Whites and have remained stagnant for the latter demographics at the four hundred sixty-eight most well-funded and selective US colleges and universities. Meanwhile, more African Americans are going into debt to fund college and fewer are graduating within six years [21].

Professional sustainability credentials, while not a panacea, offer an additional tool for bridging some of these gaps in educational attainment, employment and earnings. Beyond the STEM content and earnings links discussed above, sustainability offers other learning benefits as well, such as the ability of sustainability education to contextualize learning, as noted in a study by Jobs for the Future (JFF) discussed below [22]. When these programs include applied and work-based learning, such as through career and technical education (CTE) programs, the results are also compelling. A study of

CTE programs in Arkansas by the Thomas B. Fordham Institute, for example, found that students who take one CTE course above the national average were more likely to graduate from high school, and this was especially the case for lower income students [23].

Since most of the credentials featured here can be earned within the span of a typical semester or less, they can accelerate educational attainment and career advancement. Also, because many of the professional programs leading to credentials are offered online as complements to less frequent in-person networking and education sessions, they can help address transportation barriers. The book *Drawdown* identifies transportation as a one of the most important policy solutions for educating girls, which it lists as the sixth of 100 most effective and affordable solutions for mitigating and adapting to climate change [24].

The more extrinsic benefits of adding professional complements to academic programming are earned through active efforts by educators working with leaders of professional organizations. Examples include discounts for members, resume review, networking with experienced professionals and prospective employers and more. Academic partners can further enhance these educational and career benefits by participating as advisors on boards and committees, providing student incentives for participation, advocating for group discounts, and facilitating access in other ways. To help ensure that professional organizations have the capacity to work with increased numbers of students, higher education leaders could consider a number of strategies, including hybrid approaches such as the one, described below, that Biomimicry 3.8 has arranged with Arizona State University. A list of paginated equity references across the text can be found in Table 1 of this article to help readers link ideas about equity to the references in this article.

**Table 1.** Equitable Pathways: Summary of Benefits and Text References.

| Intrinsic Benefits | Extrinsic Benefits |
| --- | --- |
| Educational content and learning, pp. 3, 7, 8 | Membership discounts, pp. 6, 8, 9 |
| Educational access, pp. 4, 5, 8 | Credential discounts, pp. 6, 8, 9 |
| Educational attainment, pp. 4, 5, 7 | Awards and scholarships, pp. 6, 8 |
| Career advancements and compensation, pp. 3, 4, 7 | Resume review and mentoring, pp. 6, 8 |
| Professional connections, pp. 3, 8 | Member and student groups, pp. 6 |
| Social justice, pp. 5 | Featured members, pp. 6 |
| | Business opportunities, pp. 6, 9 |

## 3. Results

### 3.1. Quality Variables: Nine Indicators

When selecting among professional credentials for sustainability, the level of rigor and overall quality conveyed by the designation are obvious considerations. Quality assurances matter, especially for disadvantaged groups. The National Skills Coalition found, for example, that third-party validation can "help overcome the negative associations that employers may have regarding individuals with NDCs (non-degree credentials), making it easier for disadvantaged worker populations to enter the workforce and advance employment." [25].

Many factors go into determining quality, such as maintenance requirements and the existence of related accredited standards (and whether the credential adheres to them or requires comparable peer review). Another consideration is the degree and variety of other professional supports, such as mentoring and networking opportunities, offered by the credentialing organization. How well the credential advances actual performance for sustainability and adapts to reflect new information and practice are also tremendously important indicators. See Table 2 for a list of the nine quality indicators developed for this review.

**Table 2.** Quality Variables: Nine Indicators.

| 1. Accreditation | 4. Supports for students | 7. Member engagement |
|---|---|---|
| 2.Academic prerequisites | 5. Maintenance requirements | 8. Performance for sustainability |
| 3.Course format and content | 6. Coordination | 9. Fostering learning organizations |

### 3.1.1. Accreditation

As with the formal higher education sector, accreditation is a quality signifier for sustainability credentials from non-academic providers. Assessing these accreditations, though, is less straight forward than for the higher education sphere. Established third-party standards, such as those offered by the International Standards Organization (ISO) and its US counterpart, the American National Standards Institute (ANSI), are sometimes listed by the credential providers highlighted here. They typically list conformity to the ISO 17024 and 17027 standards for bodies certifying persons, covering such topics as impartiality, privacy, liability, examinations, maintenance, and responsiveness [26].

Yet another accreditation body, the Institute for Credentialing Excellence (ICE), founded prior to the ISO, offers the Assessment-based Certificate Accreditation Program (ACAP) through its accrediting body, the National Commission for Certifying Agencies (NCCA). The NCCA application includes an option for dual accreditation to ISO/IEC 17024 through a partnership between the Institute and the International Accreditation Service (IAS) [27]. The ICE specializes in supporting executives, staff and trustees of credentialing organizations.

Among the ICE offerings are best practice standards for accreditors in several categories: scope; organizational structure, responsibility and resources; policies and process; program records; development, delivery and maintenance; evaluation; and issuance, verification and use. Groups applying for accreditation share financial records and must typically have been in operation for at least two years.

Credentialing bodies for sustainability sometimes refer to adherence to other types of standards that do not involve accreditation. Examples include the Standards for Educational and Psychological Testing jointly developed by the American Educational Research Association (AERA), the American Psychological Association (APA), the National Council on Measurement in Education (NCME) [28] and the Next Generation Science Standards [29].

In the environmental management systems (EMS) sector, Exemplar Global provides a good example of how reference to accreditation works in practice. The organization provides certification of personnel in EMS and other quality topics. It also notes adherence to ISO/ANSI and International Electrotechnical Commission (IEC) standards as well as to the Joint Accreditation System of Australia and New Zealand (JAS-ANZ) for food schemes. Furthermore, Exemplar Global states that it also chooses accreditation of its systems from bodies that are members of the International Accreditation Forum (IAF) and that it is also part of the American Society for Quality (ASQ), a family of organizations advocating quality across commerce and trade [30].

In another configuration, the Green Business Certification Inc. (GBCI), which administers The EDGE and LEED-AP designations, but is not itself a third-party accreditation, notes that it has received and maintained personnel certification accreditation from the American National Standards Institute (ANSI) since 2011. Meanwhile, other programs, such as the Certified Biomimicry Specialist and two of the credentials offered by the Association of Energy Engineers (AEE), reference association with GBCI in various ways, but are not accredited or administered by it [31].

### 3.1.2. Prerequisites

Whether a formal post-secondary academic certificate or degree is among the prerequisites for accreditation is not only a quality consideration, but it also bears on whether a credential complements or potentially competes with higher education offerings. The Green Supply Chain Professional (GSCP)

offered through greensupplychain.org and several affiliated higher education programs, for example, list no academic prerequisite for enrollment and completion. Thus, it may be possible for a student without a high school degree to complete the program or for a student enrolled in a community college to earn the designation without completing the AA, BA or other post-secondary degree.

Having fewer academic prerequisites has both pros and cons. One of the advantages is that more individuals from lower-skilled backgrounds may be able to hasten career advancement in their fields. Applied learning, for example, may encourage high school students to graduate and pursue post-secondary education. In its study of 450 lower-skilled adults in two occupational training programs in Philadelphia and Detroit, for instance, Jobs for the Future (JFF) found that incorporating modules on water, waste, transportation, and energy, along with experiential and collaborative learning approaches, accelerated basic literacy, technical reading, applied math and computation skills [32].

Offerings that curtail or compete with formal education, however, may present disadvantages. These could include the reluctance of educators to include professional credentials or educational short-cutting that may prove costly for individuals whose earnings depend, in part, on formal educational attainment.

### 3.1.3. Format and Content

Most of the professional credentials for sustainability offer some form of study guide or curriculum, but the host's investment in the media and content vary considerably. Whether credentials are earned in person or primarily offered online and whether they are primarily self-guided or supported by online office hours and community forums can impact learning outcomes. A series of links to text, moreover, will provide a very different learning environment than audio and video content, interactive graphics and spreadsheets, virtual reality experiences, or use of artificial intelligence.

Study time and test formats vary, too. Some programs note the number of course hours and months involved in preparing for exams, while others do not. Test and recognition formats range from auto-scoring and self-printed certificates to proctored exams and staff review of applications. The USGBC's AP offerings, for example, require passing exams administered by third-party test sites, while the NWF EcoLeader Program applications are reviewed by staff before allowing self-printing of certificates or use of digital badges. The American Purchasing Society's Certified Green Purchasing Professional (CGPP) can require weeks of staff review time prior to awarding the credential. Most approaches appear to be a hybrid of self-guided work online and some level of staff review, with the option of attending on-site training sessions.

### 3.1.4. Supports for Students

A significant consideration for educators and students weighing sustainability career credentials is the degree of support offered by the industry group and staff. This can include support for the certificate content and award as well as continuing education and professional networks. Some of the online offerings include an email address for course content faculty or content developers, others list workshops and webinars, and it is worth checking the latest dates of those offerings as a way to gauge the current activity of the host. Sometimes it is difficult to find evidence of current activity. In the case of the Global Sustainability Institute of Technology (GSIT), for example, the evidence is mixed. While the last workshops listed were more than five years ago, many colleges and universities appear to be offering the content online currently.

For students breaking into diverse fields that are incorporating sustainability skills, associations often offer significant opportunities to network with professionals in the field and with prospective employers. This can include participating in local chapters, availing oneself of employer and peer mentoring webinars, volunteering in local projects, reading and publishing in trade journals, and applying for fellowships and grants. The ASCM, for example, offers a registry of individuals who have earned their professional certifications for prospective employers. The USGBC and TWS also

support student chapters and host various types of conferences and networking events. Students in these associations often receive special discounts.

### 3.1.5. Maintenance Requirements

Maintenance requirements are another indicator of the long-term value of sustainability credentials. Many hosts surveyed here require professional experience to earn a credential and continued learning to maintain it. Five years of professional experience is required by The Wildlife Society, for example, to advance from the Associate Wildlife Biologist designation to Certified Wildlife Biologist.

The Association for Supply Chain Managers (ASCM) stipulates that 70–100 professional development points must be earned every five years for maintenance of its Certified Supply Chain Professional credential. The Holyoke Community College green supply chain program and the various offerings of the Association of Climate Change Officers (ACCO), however, are among credentials that list no maintenance requirement.

### 3.1.6. Coordination

As though a race is on, multiple credential options are emerging among similar occupations and topics and collaboration among them appears so far to be rare. At least three organizations identified in Section 3 offer credentials related to carbon accounting or climate, more than one exists for clean energy, and a dozen or more exist for green building and related competencies. For those interested in general sustainability background or competencies specific to sustainability staff, at least three credentialing organizations claim to address these needs.

Where it exists, collaboration among host organizations can boost student learning, result in financial savings for participants, help differentiate among offerings, and boost confidence in the viability of the offering. In a recent example of collaboration, the Association of Climate Change Officers (ACCO) and International Society of Sustainability Professionals (ISSP) teamed up to offer a joint membership discount. In December of 2019, they also co-hosted the Global Congress for Climate Change and Sustainability Professionals in Chicago, Illinois [33].

Joining with other organizations to form groups of certificates and certifications is a recent form of coordination that may bode well for overall quality and consistency. Within the Green Business Certification Inc. (GBCI) family, for example, there are now eleven certifications for activities and places and ten accreditation opportunities for people offered by several organizations, including the USGBC [34]. In Nov 2019, the GBCI announced that it had acquired and would now administer the Sustainability Associate (ISSP-SA) and the Certified Sustainability Professional (ISSP-CSP) credentials developed by the International Society of Sustainability Professionals (ISSP) [35]. Another suite of certifications is offered by the International Living Future Institute (ILFI), which currently has certification opportunities for buildings, products, and communities, along with a professional accreditation covering all three. A third and rapidly growing cluster of sustainability certifications is offered by the aforementioned ACCO.

### 3.1.7. Member Engagement

A different type of collaboration, with members, could also help broaden audiences, improve offerings, and increase competence for sustainability across fields. One of Steve Jobs' recommendations to his protégé, Marc Benioff, was to create an application "ecosystem," in which individuals across the world could innovate and contribute to new offerings [36]. This more open-source approach to business could serve as a model for professional competencies for sustainability and is an aspect to consider when comparing offerings.

There are many ways credential hosts can adapt this approach. By allowing diverse individuals and entities to earn the right to become approved providers of training and other services, for example, the USGBC has expanded the market demand for greener buildings and accredited professionals.

The Association for Energy Engineers (AEE) is one of the approved training providers for regional efficiency programs and for the USGBC globally.

Approaches to member engagement may be one of the reasons that USGBC has more than two hundred thousand accredited professionals, whereas an organization such as The Wildlife Society has twenty-three hundred [37] even though the number of students in the US earning undergraduate degrees in both areas compare favorably (e.g. one hundred seventy thousand students earned undergraduate degrees in natural sciences and mathematics and another one hundred sixteen earned degrees in engineering in 2018) [38].

### 3.1.8. Performance for Sustainability

How well a credential advances performance for sustainability is its ultimate test. At a minimum, the "about" and "method" statements should list learning objectives consistent with the imperatives of climate science. The International Living Future Institute (ILFI) provides a good example through its list of societal imperatives, such as net zero energy and water, that guide the overall framework for certification of buildings, communities and products as well as for their accreditation of people as Ambassadors.

The ability of hosts to safeguard the integrity of its offering from special interests that might impede clean, safe renewable energy, public transport, or reduction in use of single-use plastics, as examples, is a real concern. Consulting lists of board members and trustees, mission and ethics statements, and news reviews can help discern the host's stated and actual practices.

Signaling that recipients know how to curtail new emissions to the extent and within the timetable mandated by science would be an appropriate aim for any sustainability credential. To cap global average surface temperatures to 1.5 to 2.0 Celsius, for example, will require nations to increase already ambitious planned reduction goals by three to five-fold because current targets leave a global emissions reduction gap of between 13 to 32 gigatons of carbon dioxide equivalents (GtCO2e) [39]. Credentialing groups are working in a context in which total concentrations of greenhouse gases in the Earth's atmosphere exceeded 408 parts per million (ppm) for the first time in three million years and this reality translates to a critical need for a newly skilled workforce across all sectors [40].

### 3.1.9. Learning Organizations

Effectively raising the bar to this level of performance requires credentialing organizations to engage participants in continually learning and innovating together. This involves more than technical competencies. In the twenty years leading up to development of the NWF EcoLeaders certificates, it became apparent that achieving highest performance for sustainability requires continuous learning and improvement around a range of competencies.

Therefore, it is worth evaluating whether the process of earning a particular credential fosters the communication, team-work and collegiality skills that complement technical ones. Some credentials mentioned here can be earned almost entirely online with little to no interaction with others, but the best ones offer myriad opportunities for networking, developing project management skills, co-teaching, and providing input into standards.

### *3.2. Examples*

Taking the nine quality indicators above into account, seventeen professional sustainability credentials (and fourteen additional mentions) are highlighted below. They promise to boost performance for sustainability, are career relevant and reflect aspects of good governance. They are, therefore, worth considering as complements to academic curricula. The text provides details on the relevance of each for their industries and for broader imperatives such as climate change and enhancements for educational access and learning. Category list (see Table 3).

**Table 3.** Featured Sustainability Credentials: 12 Categories of Excellence.

| | |
|---|---|
| 1. Creating high performing buildings | 7. Reporting corporate sustainability performance |
| 2. Managing energy for the future | 8. Accounting for sustainability |
| 3. Coordinating climate adaptation and mitigation | 9. Closing the loop |
| 4. Including biologists on the team | 10. Addressing the potential of tourism |
| 5. Installing clean energy systems | 11. Supporting students as leaders |
| 6. Drawing on nature for design innovation | 12. Fostering professional excellence |

It is more accurate to think of these credentials as among the best for building sustainability career competencies, rather than as educational or sustainability panaceas. Their subheadings are meant not to pigeon-hole, but to highlight some of the unique angles they present. An alphabetical list of organizations and credentials can be found in Appendix A and organizations by industry sector may be consulted in Appendix B.

### 3.2.1. Creating High Performing Buildings

Buildings account for nearly forty percent of the world's greenhouse gas emissions [41]. Add to that water, air, and health-related impacts, along with the many disciplines and occupations that intersect with buildings, and it makes sense that credentials have proliferated in this area. They vary greatly, though, in the degree to which they emphasize achieving societal imperatives such as eliminating greenhouse gas emissions and use of bio-accumulative toxins.

In the latter category is a credential offered through the International Living Future Institute (ILFI) that stands out for the aim of helping people learn how to create buildings, communities and products that give more than they take. The Living Future Accreditation (LFA) for professionals goes hand-in-hand with the ILFI standards for buildings, products and communities. Possibly the most rigorous benchmarks for buildings in the world, the ILFI standards cover seven categories called petals (place, water, energy, health and happiness, materials, equity and beauty). The imperatives related to each petal, which set forth aspirations such as achieving net-zero energy and ecological water flow, are one of the reasons the Living Future certifications and accreditation stand out.

By focusing on "critical design challenges and higher-level concerns" instead of on "basic best practices," the Institute explains, people are able to keep sights on goals that can avert a climate crisis and regenerate community. The Bullitt Center, for example, is a certified Living Building in Seattle, Washington. A six-story, 52,000 square-foot office space, the Bullitt Center is powered year-round by pollution-free sunlight for daylight and electricity, coupled with geo-exchange and passive systems for heating and cooling. The Center also harvests its own rainwater for tenants' drinking and showering needs, while recycling grey and blackwater. The net result is the building's energy use intensity (EUI) is 12.3 [42]. By comparison, the average EUI of all Seattle office buildings is approximately 92, reports Mary Adam Thomas, author of The Greenest Building [43]. Even the group of buildings meeting the LEED's highest platinum standard and Seattle's energy code, she writes, achieve an EUI of between 34 and 50.

Another pioneer in the green building space is the influential U.S. Green Building Council (USGBC) and its partner organization, the Green Business Certification, Inc. (GBCI), discussed earlier. The GBCI administers many building certification programs with accompanying accreditations for professionals, such as LEED-AP, SITES-AP, and WELL-AP. The GBCI LEED Accredited Professionals form one of the world's largest networks of professionals for environmentally responsible building trades.

Georgia Institute of Technology is pursuing both the Living Building Certification and a LEED certification at the platinum level for its new Kendeda Building for Innovative Sustainable Design. Dedicated in November 2019, the building was designed to use approximately one-third of the energy of traditional buildings in the Atlanta region, writes Josh Green, and to capture forty percent of the rainfall on the property. The building also stands out for creative reuse of materials, such as

roof shingles reclaimed for floor and wall tiles and movie sets for ceilings [44]. Students and staff participating in the design and construction of the building as well as in the current year of performance testing will be well positioned to earn the LFA or one of the LEED-AP credentials.

Earning the International Living Future Institute's (ILFI's) Living Future Accreditation (LFA) requires registering ($125), becoming a member ($50), and earning a total of 36 LFA continuing education (CE) credits. Twenty of the credits are foundational for a total fee of $525 and 16 are general credits (fees vary). The renewal fee every two years is $1000 and includes basic membership.

The introductory credential for LEED is the LEED Green Associate, followed by the LEED Accredited Professional (AP), with the option of five specialty AP certifications for building design and construction (BD + C), operations and maintenance (O + M), integrated design and construction (ID + C), neighborhood development (ND), and homes. The LEED Green Associate is a two-hour exam with 100 multiple choice questions for a fee of $250 ($200 USGBC members/ $100 students). The LEED-AP exam fee is either $550 combined ($400 for members) or $350 per specialty ($250 for members). USGBC member packages range from $100 for students and a $450 base fee for professionals to successive levels for increasing access to free course materials and other perks [45]. The USGBC member and course pages include public discussion boards in which it is possible to see inquiries about projects, courses and exams from individuals all across the world and responses by staff.

The International WELL Building Institute (IWBI) also offers an accredited professional designation (WELL-AP) administered by GBCI and an opportunity to become a member of the training faculty. Accredited professionals demonstrate latest knowledge and practice for human-centric design, especially around health and wellness in built environments. The credential is accompanied by webcasts, in-person training and various types of study guides. A network of faculty help spread the word and educate others. Although the IWBI cornerstone membership is $5000, the credential can be earned for $660–$699 depending on the bundle chosen (or for $465 for members of USGBC, or for holders of the LEED-AP or LEED Green Associate). The annual fee for WELL faculty is $500 [46].

A third accredited professional opportunity among GBCI's suite of credentials is the SITES-AP offered by the Sustainable Sites Initiative. Its purpose is to advance land practices that protect ecosystems, regulate climate, store carbon and mitigate floods [47]. The Sustainable Sites Initiative's SITES-AP program offers extensive free study materials and the exam fee is $550 ($400 for USGBC, ASLA, LEED Green Associates, LEED-APs and WELL-APs).

Also addressing responsible habitat management topics are the International Living Future Institute's (ILFI) Living Future Accreditation (LFA) and The National Wildlife Federation's Certified Habitat Steward designation [48].

### 3.2.2. Managing Energy for the Future

The Association of Energy Engineers (AEE) has more sustainability emphasis in its credentialing programs than its name conveys. In addition to a Certified Sustainable Development Professional (CSDP) credential, AEE offers designations in building commissioning, carbon reduction, demand-side management, distributed generation, energy auditing, geo-exchange design, lighting efficiency, measurement and verification, renewable energy and water efficiency.

The AEE's certified energy manager (CEM) designation is a leading global standard, recognized by the US DOE's Better Buildings workforce guidelines program as well as by the European Union's Energy Efficiency Directive (EED). The CEM is also required by many trans-national employers as a condition for hiring and helps to lend credence to the other sustainability-oriented certifications offered by AEE.

The director of sustainability for the University of California Office of the President (UCOP), Matt St. Clair, lists AEE's CEM as well as LEED-AP as his two primary professional designations on his business card and email address. St. Clair had run the energy efficiency and green building training programs for the University of California and California State University systems (thirty-three total campuses) for twelve years and chose the CEM, he explains, to help refresh working knowledge of

energy management issues. "Our campus energy managers highly valued the CEM certification," he adds, "as by far the most respected of AEE's certifications, at least in the California higher education campus facilities community." [49].

Collaboration across industry sectors for sustainability is also reflected in AEE's role as an approved training provider for the US Green Building Council (USGBC). Perks for certified members include a professional directory for employers, annual conference, ninety chapters, and many in-person and online courses scheduled throughout the year.

The thirty thousand certified AEE professionals, as of this writing, hail from 105 countries. Certification courses range from $850 for a live, online offering to $1660 for the CEM course in person, plus text fees. Members usually receive a significant course discount and many of the certification courses are offered to members only. Membership ranges from $15 for a student to $50 for a young professional and $195 for an affiliate or senior.

### 3.2.3. Coordinating Climate Adaptation and Mitigation

As nations decouple economies from carbon emissions, new jobs are emerging. The world will experience a net gain of approximately 18 million related new jobs by 2030 that help nations shift to clean energy and create circular economies, according to the International Labor Organization (ILO) [50]. Meanwhile, the ILO notes that approximately 1.2 billion jobs will be impacted by climate change, affecting vulnerable workers most severely, and requiring new adaptation and mitigation skills [51]. While all of the sustainability credentials featured here help to advance the needed skill base for these new sustainability jobs, certain roles will be instrumental for coordinating climate action initiatives, benchmarking greenhouse gas emissions goals and implementing adaptation programs that help protect the most vulnerable people and jobs.

At least three programs have emerged that provide accreditations and other perks for professionals related to climate mitigation and adaptation. They are the Association of Climate Change Officers (ACCO) Certified Climate Change Professional (CC-P), the AEE's Certified Carbon Reduction Manager (CRM), and the Urban Greenhouse Gas Inventory Specialist certification of the City Climate Planner Program administered by the GBCI.

Founded in 2008, ACCO made news when it partnered with the State of Maryland to launch the nation's first Climate Change Academy in 2018 [52]. Participating governmental and NGO organizations, that are defending lands from a rising Chesapeake Bay, are benefiting from participation in the Academy and other climate action initiatives. ACCO announced in October 2019 that one of the participants, Charles County, received a "AAA" improvement bond debt rating because of its proactive climate action steps, such as partnering with the state's Climate Change Academy [53].

ACCO is also building out a suite of new occupation-specific climate change credentials, including specializations in supply chain, energy management, financial management, planning and civil engineering, and risk management. The exams include four modules on climate science and vulnerability; greenhouse gases, energy and water management; governance, law and policy; and materiality, risk management and economics. The course content includes a mix of live, online training and seventeen illustrated slide decks.

Slide decks are available for between $20 and $40 each and including 50–100 slides. The ACCO CC-P exam and application bundle are $695 for non-members and $495 for members (content is available separately). Information about accreditation and renewal requirements is not listed. The ACCO membership is $125 annually for government employees and $195 for general professionals.

AEE's Certified Carbon Reduction Manager (CRM) exam is $300 if taken with the live seminar for a fee that ranges from $850 to $1660. AEE offers discounts for members which costs $15 for students, $50 for young professional, and $195 for affiliates and seniors.

Urban Greenhouse Gas Inventory Specialist certification of the City Climate Planner Program administered by the GBCI requires an application fee of $50, exam fee of $400, and renewal every five years for $100. An online course is offered a la carte via the USGBC for $99.

### 3.2.4. Including Biologists on the Team

Professional certifications in the biological sciences, wildlife biology, or natural systems may, at first, seem narrower than the broader goals of climate action or sustainability. A closer look, however, reveals that these designations provide depth in the three primary aspects of sustainability: ecosystem services, ecological footprint, and solutions. A good case can be made that any team for sustainability should include professionals who have a background in biodiversity, natural systems and limits. This is true not only because of accelerated loss of species diversity and abundance all across the world [54], but also because nature can inform solutions to correct the balance.

The Wildlife Society's Associate Wildlife Biologist (AWB®) and Certified Wildlife Biologist (CWB®) are among the strongest credentials for those who want to bring an understanding of the interplay of natural and social sciences to sustainability teams. In addition to rigorous post-secondary academic and experience requirements for earning either of its two possible professional designations, successful applicants must hold a minimum Bachelor of Sciences or Arts in related fields, although specific course work together with experience can sometimes substitute for the four-year degree.

The number of semester hours required by The Society in social sciences (e.g., communications, law, policy, etc.) is approximately equal to the number required in the biological and physical sciences. In addition to conveying that members meet minimum educational, experience and ethical standards, explains The Society, the credentials signal the "high technical and social skills needed by today's wildlife biologists." [55].

Reflecting wider applicability than is conveyed by the organization's name, the jobs posted on The Society's website vary considerably. They include such positions as Director of Inclusion and Diversity in the natural resources program at Virginia Tech, as well as postings in agriculture and natural resources, forest ecology, raptor and ecosystem health and wetland monitoring.

Given that more than one hundred ten thousand students in the US graduated in 2015–2016 from degree programs in agriculture, natural resources, and biological sciences, there is great room to expand professional certification in allied fields [56]. As of 2019, The Society supports four hundred Associate and one thousand nine hundred Certified Wildlife Biologists, a small and elite minority within the fields of wildlife and natural science.

Enhancing its professional certifications are The Society's annual conferences, student chapters, magazines and a bulletin, as well as an online directory of certified members with email access for prospective employers. Life-long learning is supported by The Society's five-year certification maintenance requirement as well as through the additional offering of a professional development certificate. The fee for the AWB® is $115 and for the CWB® $155 (plus a $25 maintenance fee every five years for the latter).

### 3.2.5. Installing Clean Energy Systems

Competitive with the AEE's programs, the North American Board of Certified Energy Practitioners (NABCEP) provides accreditations for several renewable energy roles. One distinction is that, whereas the AEE's most distinguished credential, the Certified Energy Manager (CEM), encompasses overall building energy management, the NABCEP focuses on more specialized skills related to renewable installations, sales and inspections (albeit AEE offers several of these as well).

Specifically, NABCEP offers certifications in photovoltaic, solar heating and small wind installation. Other NABCEP designations focus on competencies in design, inspection, commissioning and maintenance. Minimum experience includes having had a major decision-making role in design and installation of a minimum 1 kW system that has gone through permitting and final inspection as well as minimum training through the Occupational Safety and Health Administration (OSHA). Applicants must also be at least eighteen years old.

Johnson County Community College's sustainability director and history professor, Jay Antle, notes that the NABCEP certifications are ones that students sometimes pursue in conjunction with the colleges' various environmental technology and sustainability degree programs [57].

Most of the NABCEP professional certifications require an application fee ($125) and exam fee ($375). The PV system inspector (PVSI) and solar heating inspector (SHI) credentials cost a bit less (no charge for registration and $150 for exam). The three introductory NABCEP Associate credentials include application ($25) and exam fees ($125). Re-certification is required every four years, with applications due in the third year. Testing services require an additional $75 for exams taking place outside of the US.

### 3.2.6. Drawing on Nature for Design Innovation

How nature can inform design for sustainability is the basis of this professional credential that is now offered as a Master's Certificate by Arizona State University. As the co-founder of the program and author, Janine Benyus, states, "Biomimicry ushers in an era based not on what we can extract from nature, but on what we can learn from her. This shift from learning about nature to learning from nature requires a new method of inquiry." [58].

Starting as an online course offered by a certified B-Corp, the credential recently migrated to an online platform of a large accredited university. The course is also approved by Green Business Certification Inc. (GBCI), and engages professionals from diverse disciplines, ranging from education and engineering to planning and product design. The immersion workshops create a diverse cohort of professionals who can continue to collaborate with support of an online community called the "Reef" hosted by the non-profit organization, Biomimicry 3.8.

A series of five classes and related assignments, totaling fifteen credit hours offered over fifteen weeks, fulfill one of two requirements for certification. The other requirement is participation in two immersion workshops offered in diverse bioregions across the world, ranging from the tropical rainforests and mangroves of Costa Rica to the Savannah grasslands of South Africa.

The fee for the online portion of the course is $11,400 at this writing ($760 per credit hour) and the immersion workshop fee ranges from $3000 to $5000 each plus travel. A longer program with more requirements leads to a Master's of Science from Arizona State University and designation as a Certified Biomimicry Professional.

### 3.2.7. Reporting on Corporate Sustainability Performance

When the Exxon Valdez tanker spilled 11 million gallons of oil into the delicate Prince William Sound ecosystem on March 24, 1989, it launched a new wave of corporate responsibility initiatives, including the CERES Principles of the Coalition for Environmentally Responsible Economies. In 1997, the Global Reporting Initiative (GRI), based in the Netherlands, aimed to create an international standard that would unify myriad corporate sustainability reporting initiatives [59].

Demonstrating understanding of the reporting scope and requirements, individuals can earn a certificate of achievement by taking an online course and passing related exams covering the basics of reporting, including modules on the economic, environmental and social dimensions of sustainability. A network of certified training partners across the world, including Boston College, BSD, Deloitte, ERM, ISOS group, Keramida, and RWDI, in the US, provide support with preparing and registering GRI reports and with related professional training opportunities. One of the providers, Boston College, for example, offers an on-site accelerated GRI reporting certificate summit [60]. The next one is on April 30, 2020. GRI offers a registry of individuals who have earned the certificate of achievement.

The online certificate course fee is $1350 for members and $1550 for non-members and the exam fee ranges from $190 to $300. Annual membership fees are pegged to organizational revenues and range from $500 for organizations with less than one million dollars in revenues to $22,000 for organizations with more than 50 billion dollars in revenues.

### 3.2.8. Accounting for Sustainability

Company leaders, financial analysts, and investors want to know which of hundreds of sustainability impacts and topics are most likely to influence net returns and overall performance for

specific sectors, industries, and companies. Several corporate reporting initiatives, including GRI and the Task Force on Climate Related Financial Disclosures (TCFD), assist finance leaders and investors in assessing sustainability materiality and risk [61]. [It should be noted that the ACCO will also soon offer additional accreditations in financial management (CC-FM) and risk management (CC-RM)] [62].

Among these resources for sustainability assessment, The Sustainability Accounting Standards Board (SASB) is unique for assessing finance materiality specific to each industry and major sector. In 2018, SASB approved a set of seventy-seven standards encompassing eleven sectors. A Sustainability Industry Classification Systems (SICS), that mirrors traditional labor market classifications, guides the SASB financial standards [63].

The SASB sectors are: consumer goods, extractive and mineral processing, financials, food and beverage, healthcare, infrastructure, renewable resources and alternative energy, resource transformation, services, technology and communication and transportation. To give one example, SASB standards in the consumer goods sector cover each of the following business categories: apparel, accessories and footwear, appliance manufacturing, building products and furnishings, e-commerce, household and personnel products, multiline and specialty retailers and distributors, and toys and sporting goods [64].

Already more than one thousand individuals have registered for or earned SASB's Fundamentals of Sustainability Accounting (FSA) credential, offered as an assessment-based certificate. Test takers range from the Senior Supply Chain Analyst for Gap to the Chief Investment Officer for the California State Teachers' Retirement System [65]. Earning the credential requires taking and passing two exams: level one (on principles and practices) and level two (on application and analysis). For members, the fee for levels one and two is $900; for non-members, it is $1100. Membership is $400 for individuals and $200 for educators. SASB does offer engagement opportunities and member listings, but does not state third-party accreditation, maintenance or other requirements for earning the assessment-based certificate.

### 3.2.9. Closing the Loop

Supply chain management (SCM) is a well-established field that, traditionally, has aimed to improve companies' competitiveness by lowering costs and speeding production. It is a complex and multi-faceted endeavor, encompassing most of the activities across a product's life cycle, from sourcing to manufacturing and transportation, to packaging, use and reclamation [66].

Sustainable SCM can be a powerful driver for social responsibility across industries, occupations and regions. Goals include reducing climate-altering emissions and other forms of pollution, addressing human rights, and improving public health. Tactics for achieving those goals can include shortening and integrating supply chains [67], shifting to renewable and safe forms of energy, digital labeling (or blockchain) to improve communication and monitoring, and closed loop manufacturing. Interface, Inc., for example, collects its modular carpet squares and Eileen Fisher reclaims used clothing for reuse in manufacturing.

As SCM for sustainability gains traction, an array of certification options is emerging. These include higher education credentials and degrees offered at colleges and universities across the world. Other options range from a free online primer to credentialing offered through a professional association that set early standards for SCM and is beginning to include sustainability considerations, to a sustainability institute that offers diverse specialized certificates.

For an orientation to the topic, a free online tutorial, "Supply Chain Sustainability" is offered through Open University [68]. It requires approximately eight hours to complete.

The industry standards for SCM were developed by APICS, which recently launched the Association for Supply Chain Managers (ASCM). In addition to a magazine, employer verification program and educational events all across the world, the ASCM offers a series of certifications. Sustainability is listed as an aspect of one of three modules for the Certified Supply Chain Professional (CSCP) [69].

Like the HCC's SCSP option, the ASCM's CSCP is a self-guided and paced program, but the prerequisites are higher and sustainability is not the core focus. The fee for the course and materials is $1385 for core members. Core membership is $180 per year at this time. Maintenance requires earning 70–100 qualifying professional development points every five years. The prerequisite is a minimum Bachelor's degree or one of several other related certifications offered by the association.

Holyoke Community College (HCC) [70] is one of several colleges and universities across the US offering the Certified Green Supply Chain Professional (SCSP) designation online. HCC notes that after taking this course, students are qualified to apply for either the Senior Certified Sustainability Professional (SCSP) or the Green Supply Chain Professional Certification (GSCP) through greensupplychain.org [71]. The latter organization indicates on its website that 2000 colleges and universities across the world currently use its certification programs for their courses [72]. Topics include purchasing, carbon reducing strategies, sustainability 101 and transportation. The fee for the self-guided, self-paced program ranges from $1595 to $1600.

Note also that the Sustainable Purchasing Leadership Council (SPLC) offers a vibrant community and awards program that often features supply chain topics in its webinars and conferences but does not currently offer a credential. Among its free online resources are presentations on trends in supply chain disclosure and traceability, including use of software programs such as Sourcemap [73]. The association, ACCO, discussed above is also developing an accreditation focused on reducing carbon pollution across supply chains, the CC-PSP [74].

### 3.2.10. Addressing the Potential of Tourism

Overshadowing its potential to help lift people out of poverty, tourism has recently been recognized as a considerable source of global greenhouse gas pollution. The footprint is approximately 8 percent of global emissions, reports Marina Kelava for the Ecologist, which is more than four-times larger than previously estimated [75].

Several credentials aim to help professionals mitigate the impacts of tourism and seize the opportunity for good. Universities such as Arizona State University, George Washington University, New York University and the University of North Carolina Greensboro, for example, offer certificate and degree programs in ecotourism and sustainable tourism. Non-governmental organizations have also played a role. Until 2018, The International EcoTourism Society (TIES) founded in 1990, marketed an annual series of conferences on responsible tourism as well as three certificate programs. Website information indicates that its accreditation programs were transferred recently, though, to George Washington University.

Meanwhile, the Global Sustainable Tourism Council (GSTC), based in Switzerland, has stepped in as a sustainable tourism accreditation body. In addition to certification for destinations, the UN-endorsed GSTC offers a Certificate in Sustainable Tourism that can be earned by passing an exam within a month after attending either one of the in-person, three-day classes held throughout the world (in 2020, courses are scheduled in Indonesia, Japan and Portugal) or an online course. The latter is accompanied by weekly video lessons, hands-on exercises, and interactive and collaborative sessions. The GSTC individual membership is $100 and in-person classes cost between $300 and $370. Pricing for the online course is not stated. A member only section and listing of member organizations is available on the site. Experience, education and renewal requirements are not listed.

### 3.2.11. Supporting Students as Leaders

Plan, do, check, act. This idea has long roots among business leaders in total quality management (TQM) circles. The International Standards Organization (ISO) environmental management standards, for example, are built upon this continuous improvement cycle. It adapts to support student learning and agency for sustainability as well, avoiding prescriptive menus of action in favor of local adaptation.

Branching from this idea, for example, NWF's *EcoLeaders Program* bases its student credentials on a continuous learning and improvement model. Earning either the NWF Certified *Campus* or *Community*

*EcoLeader* credentials requires an active role in sustainability project design or implementation, a self-assessment reviewed by staff on the program's core learning objectives, and reported efforts in the four learning areas: planning, doing, communicating, and sustaining.

Leaders work through a five-step program of defining purpose, developing project plans, supporting others, earning either a campus or community certificate for project leadership, and developing career paths. The credentials encompass one or more of thirteen topics: buildings, campus and community policy, climate change, community and environmental justice, consumption and waste, education and awareness, energy, food, habitat and wildlife, outdoor recreation and leadership, purchasing, transportation and water. Among the student supports are a career center, online community, conference and webinars and a variety of action-specific badges that promote such campaigns as starting an EcoSchool or creating wildlife corridors.

The fee for earning either EcoLeader Credential is $30 and includes free access to the career planning tool and other resources.

3.2.12. Fostering Professional Excellence

Many of the credentials above concentrate on specific industries and the occupations within them, such as business and finance or building and construction. An ongoing need is to coordinate these activities in a way that the whole is greater than the sum of its parts. The International Society of Sustainability Professionals (ISSP) addresses this particular need.

Supporting the unique challenges of those whose task is to coordinate sustainability activities across departments and communities, ISSP plays a key role in establishing norms and advancing best practices across the field. The organization offers a regular series of webinars on topics ranging from embedding sustainability across cultures, to communicating sustainability expertise and addressing food security [76]. With ACCO, ISSP has recently co-hosted a global congress for climate leadership and regularly collaborates with other organizations [77].

The ISSP, acquired by GBCI in November of 2019 [78], offers two credentials, the Sustainability Associate (ISSP-SA) and the Certified Sustainability Professional (ISSP-CSP). Although anyone can take the ISSP-CSP exam, earning the designation requires five years of full-time sustainability related work experience, and applicants are required to take the ISSP-SA first. As with many of the credentials explored here, exams are taken at testing centers or via remote proctoring in real time. Credential renewal is every two years and requires continuing education credits. The ISSP has offered discounts for Education Partners including AASHE, ACCO and USGBC.

The ISSP-SA was launched in 2016 and has 240 current credential holders while the ISSP-CSP was launched in 2017 and currently has 50 credential holders. The ISSP-SA exam fee is $350 ($175 for members. Renewal is $150 ($75 for members). The ISSP-CSP application fee is $150 ($75 for members) and the exam fee is $300 ($175 for members). Renewal is $200 ($100 for members). Membership is currently $150 and includes an active roster of webinars, networking opportunities, job postings and a credential holder database. ISSP has offered discounts for education partners such as AASHE and ACCO and a joint membership opportunity with ACCO. Another established source of knowledge for sustainability professionals is greensupplychain.org, which, along with the Green Supply Chain Professional credential, offers the Certified and Senior Certified Sustainability Professional designations [79].

## 4. Discussion

These credentialing efforts collectively point to a growing recognition that sustainability skills are an imperative for society. As the twenty-four million new middle- and higher-skill jobs identified by the International Labor Organization come into being over the next 10 years, marginalized communities and students stand to gain, if they are well positioned. Although it is often standard procedure for community colleges to embed professional certifications into academic programming, writes Goldie Blumenstyk in the Chronicle of Higher Education, four-year liberal arts colleges and universities have

been slower to incorporate them [80]. For both levels of higher education, sustainability credentials are relatively new.

*4.1. Ideas for Integration*

For faculty, academic deans and other education leaders contemplating whether and how to offer professional sustainability credentials along with the formal curriculum, these seven ideas draw on the lessons above:

1.  Ask for help with research: Staff at many sustainability credentialing organizations would be glad to organize an introductory seminar. Student groups from student affairs to sustainability organizations may be willing to help with research and even fundraising to raise student member, application and test fees. Career services, campus research, academic and alumni affairs may be other sources of support.

2.  Consider imperatives: The science is clear that a near elimination of carbon emissions is necessary across the world by 2050 to secure ecosystem services for current and future generations. This equates to bigger shifts in practices related to restoration of habitat, energy use, built environments and infrastructure than is currently being signaled across society, even by environmental organizations. Sustainability credentialing organizations vary as well in their presentation of these top imperatives. This is a factor to consider, as faculty and students determine how to engage with sustainability credentialing organizations.

3.  Reward collaboration: A race is on to 'credentialize' sustainability, potentially creating confusion and silos. On the plus side, diversity in credentialing makes room for new leadership and ideas and may raise the bar for performance overall. Nevertheless, efforts to collaborate should be encouraged. Credentialing groups may at least acknowledge and stay abreast of one another's work or offer joint conferences, discounts and even standards. Faculty and students can help by either choosing credentialing organizations that show evidence of active collaboration or by asking them to do so.

4.  Look for accreditation: The credentials surveyed here varied widely in their relationship with third-party accreditations. Because credentialing itself is an established field with unique responsibilities, liabilities and best practices, it is helpful for staff of sustainability credentialing organizations to be a part of them. This can also be a quality assurance for faculty.

5.  Experiment across disciplines: Although most of the credentials featured here apply to specific industries, the competencies they represent are relevant for most any teams working on sustainability projects of any type or size. Considering the typical phase of projects, for example, points to the need for sustainability competencies in planning, natural systems, project coordination and management, financial management, legal and regulatory understanding, purchasing, technical implementation, commissioning, communications, education, maintenance, assessment and reporting in a continuous feedback loop of learning and improvement. There is an emerging body of knowledge around competency clusters for sustainability that can provide insight into how to formulate teams [81]. Faculty can help by encouraging students to participate in myriad sustainability credentialing opportunities and to apply and share these ideas in project-based learning experiences on campus or in the wider community.

6.  Solicit help with fundraising: Even though most fees for sustainability credentials are relatively low compared with industry norms, one of the barriers for many students will be the application, exam, membership and maintenance fees for earning sustainability credentials. Test study materials and some travel may also be involved. College administrators can help by asking credentialing organizations to offer student and group discounts, as many of them already do. Local businesses may be willing to offer scholarships or other types of funding to help boost students' career readiness in their industries. It is also possible to shop for credentials based on

affordability. Some, such as the TWS credentials and the International Living Future Institute's LFA, are a particular bargain.

7.  Provide student supports: Not all of the sustainability credentials are created equally when it comes to active supports for students. Some are particularly geared to help students network with one another and with more seasoned professionals and employers across fields. This is another important factor to consider when deciding whether or not and how best to enhance options.

### 4.2. Areas for Future Research

This survey article points to questions that could occupy many students. Categories for more research are numerous. For example, one topic is societal imperatives and the degree to which credentials help accomplish them or scatter priorities. Another is equity and inclusion and identifying the best practices for ensuring students from lower-skilled backgrounds are introduced to these opportunities to distinguish themselves in some of the best occupational opportunities in the world. Studies to compare differences in academic and career attainment for students who obtain professional credentials for sustainability would be complex, but well worth the effort.

Another topic might be how credentials are drawing on nature for design inspiration and extending beyond what might be required by US National Environmental Policy Act (NEPA) and Endangered Species Act (EPA). Some of the credentials appear to pay little heed to biodiversity and ecosystem service, such as the ratio of built to green space, stream and river corridors and buffers and could be enhanced in those ways. Then there are the gaps in topics encompassed by credentials. Some topics such as food, wood and other goods, as well as transportation, are beginning to be addressed by sustainability certifications for goods, places and services, but do not yet recognize people. The emergence of and need for new credentials, thus, will be another area for research, study, and entrepreneurship. The politics and internationalization of these efforts, moreover, as well as composition of boards and staff and overall sustainability profile of the organizations themselves, will all become more relevant with time as well.

### 5. Conclusions

Sustainability is becoming embedded across sectors and industries as an occupational standard. It is a sophisticated pursuit with high stakes for people and planet. Professionalization of sustainability conveys recognition by experts across industries and occupations that people and planet matter. Overall, this trend is positive and can be a driver for increased inclusion, equity, job growth and demand for updated skills across fields.

The trend presents a window of unique opportunity for individuals to distinguish themselves and advance as new standards for performance take hold. In large fields in which relatively few have earned the professional credentials for sustainability, individuals from under-served backgrounds have a particular opportunity to make connections, improve standing, increase visibility, seize competitive positions, raise the bar for occupational performance and gain the market insights necessary to start successful new businesses and organizations or to improve existing ones. Seizing and shaping these trends can ultimately strengthen and protect people and our planet.

**Funding:** This research received no external funding.

**Acknowledgments:** The author would like to thank Krista Hiser, a national leader for sustainability curricula, for critique, discussion, ideas and support in the conceptual and research phases of this article as well as Association for Advancement of Sustainability in Higher Education (AASHE), the National Council for Science and Environment (NCSE) and the Sustainability Curriculum Consortium (SCC) for supportive forums, resources and webinars. Also, a thank you to Fritz Myer, Aaron Allen, April Keniry and Tina Evans, and anonymous reviewers, for reading drafts and to Jay Antle, Matt St. Clair and John Kanter for providing insights into how they are using credentials for sustainability.

**Conflicts of Interest:** The author declares no conflict of interest.

# Appendix A

**Table A1.** Alphabetical Listing of Organizations and Credentials.

| Organizations, Industry and Trade Associations (Alphabetical Order) | Credentials ** |
|---|---|
| **Association of Climate Change Officers (ACCO)**<br>Certified Climate Change Professional (CC-P)<br>Certified Climate Change Officer (CC-O)<br>City-County Management (CC-CM)<br>Energy Management (CC-EM) | Financial Management (CC-FM)<br>Planning and Civil Engineering (CC-PCE)<br>Risk Management (CC-RM)<br>Supply Chain and Procurement (CC-PSP) |
| **Association of Energy Engineers (AEE)**<br>Building Commission Professional (CBCP)<br>Business Energy Professional (BEP)<br>Carbon Reduction Manager (CRM)<br>Certified Energy Auditor (CEA)<br>Certified Sustainable Development Professional (CSDP)<br>Certified Building Energy Simulation Analyst (BESA)<br>Certified Energy Manager (CEM)<br>Certified Residential Energy Auditor (CREA)<br>Demand Side Manager (CDSM),<br>Distributed Generation Professional (DGCP) | Energy Efficiency Practitioner (EEP)<br>Energy Procurement Professional (EPP)<br>Existing Building Commissioning Professional (EBCP), Geothermal Designer (CGD)<br>Industrial Energy Professional (CIEP)<br>Lighting Efficiency Professional (CLEP)<br>Measurement and Verification Professional (CMVP)<br>Performance Contracting and Funding Professional (PCF)<br>Power Quality Professional (CPQ)<br>Renewable Energy Professional (REP)<br>Sustainable Development Professional (CSDP)<br>Water Efficiency Professional (CWEP) |
| **Association for Supply Chain Managers (ASCM)** | Certified Supply Chain Professional (CSCP) |
| **Biomimicry 3.8 and Arizona State University and its Biomimicry Center *** | Graduate Certificate in Biomimicry (Biomimicry Specialist Certification)<br>Master of Science (MS) in Biomimicry (Biomimicry Professional Certification) |
| **City Climate Planner Program/GBCI** | Urban Greenhouse Gas Inventory Specialist |
| **Global Reporting Initiative** | Certified Training Partner<br>GRI Certificate of Attendance |
| **Global Sustainable Tourism Council (GSTC)** | Certificate in Sustainable Tourism |
| **Greensupplychain.org** | Certified Green Supply Chain Professional (GSCP)<br>Senior Certified Sustainability Professional (SCSP) |
| **Green Business Certification, Inc. (GBCI) *** | USGBC LEED-AP and others * |
| **International Institute for Sustainability Professionals (ISSP)** | Sustainability Associate (ISSP-SA)<br>Certified Sustainability Professional (ISSP-CSP) |
| **International Living Future Institute (ILFI)** | Living Future Accreditation (LFA) |
| **International WELL Building Institute (IWBI)** | WELL-AP |
| **National Wildlife Federation** | Certified EcoLeaders (Campus and Community)<br>Certified Habitat Stewards |
| **North American Board of Certified Energy Practitioners (NABCEP) Board Certifications**<br>The PV Commissioning and Maintenance (PVCMS)<br>The PV Design Specialist (PVDS)<br>The PV Installation Professional (PVIP) | The PV Installer Specialist (PVIS)<br>The PV System Inspector (PVSI)<br>The PV Technical Sales (PVTS)<br>The Solar Heating Installer (SHI)<br>The Solar Heating System Inspector (SHSI) |
| **Sustainability Accounting Standards Board (SASB)** | Fundamentals of Sustainability Accounting (FSA) Certificate |
| **Sustainable Sites Initiative** | SITES-AP |
| **U.S. Green Building Council (USGBC)/GBCI** | LEED Green Association, Accredited Professional (LEED-AP), Building Design and Construction (LEED BD+C), Operations and Maintenance (LEED O+M), Integrated Design and Construction (LEED ID+C), Neighborhood Development (ND) and LEED Homes |
| **The Wildlife Society** | Associate Wildlife Biologist (AWB) and Certified Wildlife Biologist (CWB) |

* As detailed in the text, GBCI administers several professional organization's sustainability credentials and is associated with the USGBC. ** Note also that most of the organizations have trademarked their credential names. *** Arizona State University hosts the credentials, but the founder is Biomimicry 3.8 and various resources are offered in partnership, as discussed in the text; Those organizations, industry and trade associations with bold and underline are the association headers.

## Appendix B

**Table A2.** Featured Professional Organizations and Industry Sectors (a).

| Professional Organizations, Industry and Trade Associations (b) | Represented Industry Sectors |
|---|---|
| Association of Climate Change Officers and the City Climate Planner Program | Administrative and Support and Waste Management Services; Professional, Scientific and Technical Services; Other Services |
| Association of Energy Engineers (AEE) | Utilities; Professional, Scientific, and Technical Services; and Public Administration |
| Association for Supply Chain Managers (ASCM) and Greensupplychain.org | Public Administration; Professional, Scientific, and Technical Services |
| Biomimicry 3.8 and Arizona State University (c) | Professional, Scientific, and Technical Services |
| International Institute for Sustainability Professionals (ISSP) | Professional, Scientific, and Technical Services |
| International WELL Building Institute, Living Future Institute, and the U.S. Green Building Council (USGBC) | Construction; Manufacturing; Professional, Scientific, and Technical Services; Utilities |
| Global Reporting Initiative | Public Administration; Professional, Scientific, and Technical Services |
| Global Sustainable Tourism Council (GSTC) | Administration and Support and Waste Management and Remediation Services; Accommodation and Food Services |
| The National Wildlife Federation and The Sustainable Sites Initiative | Agriculture, Forestry, Fishing, and Hunting; Other Services |
| The Wildlife Society | Professional, Scientific and Technical Services |
| The North American Board of Certified Energy Practitioners (NABCEP) and Association of Energy Engineers (AEE) | Construction; Professional, Scientific, and Technical Services |
| The Sustainability Accounting Standards Board (SASB) | Public Administration; Professional, Scientific, and Technical Services |

(a) Using the North America Classification System (NAICS) of industries, http://www.naics.com/search/. (b) Alphabetic order and grouped by topic. (c) Arizona State University is the academic host in collaboration with Biomimicry 3.8.

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
