# Peer review of "Equitable Pathways to 2100: Professional Sustainability Credentials"

_sustainability, doi:10.3390/su12062328_

Round 1

Reviewer 1 Report

This draft paper is a review which is well writing but the abstract does not represent the paper or at least there in not good connection:

a) line 8. A table or a list with the industries and profesional association should be made along the paper

b) line 14. A table or list with the seventeen professional sustainability credentials and the fourteen additional mention should be made along the article.

b) line 18. A  table or list t of aspect to considered and new aspects should be made along the article

c) line 20 and 21. A table or list of no academic and  professional credential should be made along the article

d) line 22. A table or list with the set of suggestions should be made along the article. 

Alsto, in the line 824 supplementary material  does not appear in the draft paper. if there is not, it should be eliminated. It is confusing

Author Response

February 18, 2020

Thank you for your time and feedback.  They are invaluable.  As you suggested, I am in the process of making these edits:

  1. Creating tables and/or lists of associations, credentials and suggestions.
  2. Strengthening links with the abstract and/or revising the abstract accordingly.

Once edited, the paper will be resent accordingly, with tracked changes.  

I hope that your work for sustainability goes very well and that our paths will cross in the future.

Very best regards,

L. Julian Keniry

+1-202-999-9244

Reviewer 2 Report

The paper was well written and should be of interest to readers.

Minor edits:

1.  Lines 96, 111, 343--quotes should be "

2.  Line 200--consideration instead of considerations

3.  Line 227--no comma after Global

4.  No comma after administers

Author Response

February 18, 2020

Thank you for your feedback and time.  They are invaluable.

I am in the process of making the edits to the lines you suggested and greatly appreciate your specificity.

Wishing you all the very best in your work for sustainability and that our paths may cross someday.

Best regards,

L. Julian Keniry

+1-202-999-9244

[email protected]

This manuscript is a resubmission of an earlier submission. The following is a list of the peer review reports and author responses from that submission.